# How Do Decision Makers and Service Providers Experience Participatory Approaches to Developing and Implementing Physical Activity Interventions with Older Adults? A Thematic Analysis

**DOI:** 10.3390/ijerph18042172

**Published:** 2021-02-23

**Authors:** Amy Davies, David P. French, Angela Devereux-Fitzgerald, Elisabeth Boulton, Chris Todd, Chris Phillipson, Laura J. McGowan, Rachael Powell

**Affiliations:** 1Manchester Centre for Health Psychology, Division of Psychology & Mental Health, School of Health Sciences, University of Manchester, Manchester M13 9PL, UK; amy.davies@manchester.ac.uk (A.D.); Angela.Devereux@manchester.ac.uk (A.D.-F.); rachael.powell@manchester.ac.uk (R.P.); 2Manchester Academic Health Science Centre, Manchester M13 9PL, UK; elisabeth.boulton@manchester.ac.uk (E.B.); Chris.Todd@manchester.ac.uk (C.T.); 3National Institute for Health Research, Applied Research Collaboration-Greater Manchester, Manchester M13 9PL, UK; 4Division of Nursing, Midwifery & Social Work, School of Health Sciences, University of Manchester, Manchester M13 9PL, UK; 5School of Social Sciences, University of Manchester, Manchester M13 9PL, UK; christopher.phillipson@manchester.ac.uk; 6Population Health Sciences Institute, Newcastle University, Newcastle Upon Tyne NE1 7RU, UK; laura.mcgowan@newcastle.ac.uk

**Keywords:** physical activity, older adults, co-production, co-design, place-based, engagement, acceptability, sustainability, healthy ageing, intervention

## Abstract

Background: Physical activity has numerous health and well-being benefits for older adults, but many older adults are inactive. Interventions designed to increase physical activity in older adults have typically only produced small effects and have not achieved long-term changes. There is increasing interest in participatory approaches to promoting physical activity, such as co-production, co-design and place-based approaches, but they have typically involved researchers as participants. This study aimed to understand the experiences of decision-makers and service developers with the introduction of such participatory approaches when developing new physical activity programmes outside of a research setting. Methods: Semi-structured, qualitative interviews were conducted with 20 individuals who were involved in commissioning or developing the Greater Manchester Active Ageing Programme. This programme involved funding eight local authorities within Greater Manchester, England, to produce physical activity projects for older adults, involving participatory approaches. An inductive thematic analysis was conducted, structured using the Framework approach. Results: Interviewees identified important benefits of the participatory approaches. The increased involvement of older adults led to older adults contributing valuable ideas, becoming involved in and taking ownership of projects. Interviewees identified the need to move away from traditional emphases on increasing physical activity to improve health, towards focussing on social and fun elements. The accessibility of the session location and information was considered important. Challenges were also identified. In particular, it was recognised that the new approaches require significant time investment to do well, as trusting relationships with older adults and partner organisations need to be developed. Ensuring the sustainability of projects in the context of short-term funding cycles was a concern. Conclusions: Incorporating participatory approaches was perceived to yield important benefits. Interviewees highlighted that to ensure success, sufficient time needs to be provided to develop good working relationships with older adults and partner organisations. They also emphasised that sufficient funding to ensure adequate staffing and the sustainability of projects is required to allow benefits to be gained. Importantly, the implementation of these approaches appears feasible across a range of local authorities.

## 1. Introduction

Physical activity confers various benefits to older people, including improved well-being, a reduced illness risk, and an increased life-expectancy [1]. However, older adults are the least physically active age group and activity declines with advancing age [2]. In England, in 2016, 44% of adults aged ≥ 65 years engaged in 150 min of moderate intensity physical activity a week, compared with 67% of adults aged 19–64 years [3]. 

A number of interventions have been designed to promote physical activity in older adults. These can be effective in increasing activity up to one year later [4], but the activity increases are generally small, and typically smaller than those produced by interventions with younger age groups [5]. Furthermore, these increases are not apparent beyond one year [4]. One possible explanation for this lack of maintenance is that many older adults take part in physical activity programmes to increase social contact and to take part in fun activities, rather than through a desire to increase their physical fitness [6]. However, many interventions do not aim to meet these older adults’ need for social contact and enjoyment [6]. Additionally, qualitative studies of older adults who were not taking part in interventions to increase their physical activity have revealed indifference or even hostility to the idea of increasing physical activity for its own sake [7]. In sum, individually delivered interventions to promote physical activity in older adults produce small effects that are often not maintained, and may be of limited interest to many older adults. 

There is increasing interest in participatory approaches to promoting physical activity [8]. These centre around promoting the participation of older adults in the development of physical activity interventions, which are valued in the locations they are implemented, in order to increase physical activity. Such approaches typically aim to embed long-term, sustainable physical activity programmes within the neighbourhoods where older adults live, rather than being ‘interventions’ delivered for a fixed period of time and then withdrawn. 

The present research considers various participatory approaches to involving older adults, including co-production, co-design, place-based working, and an asset-based approach. There are a variety of definitions for these approaches that derive from different disciplinary backgrounds [9], leading to frequent areas of disagreement [10,11]. In the present research, terms are used as follows: Co-production is an umbrella term for activities that aim to fully involve end-users in the development of interventions, by viewing the experiential knowledge of these end-users as core to the success of their development [10]. A related concept—co-design—emphasises involvement in identifying the problem and how to go about addressing it, rather than involvement in the development or delivery of interventions [12]. A place-based approach considers both local needs and local assets [13], drawing on older people’s extensive knowledge of the communities and environments in which they live. This relates to taking an asset-based approach, where the experiences and skills of older adults are recognised and valued, as are the resources available in a local area [14]. 

Despite the growing interest in using these new approaches, there have been few evaluations of their success, especially in relation to groups such as older adults [15]. In particular, there is limited evidence concerning the effectiveness and impact of participatory approaches with older adults [15]. There has been some evaluation of such programmes, aimed at older adults and adults with health conditions, delivered in partnerships by the local government, local NHS organisations and voluntary sector organisations. These suggest increased levels of physical activity in those who persist with the programmes, as well as improved mental wellbeing, with some projects continuing beyond the funding period [16,17]. Furthermore, many existing co-production exercises have involved researchers as a key participant group alongside service providers and service users [18]. There is limited knowledge about the effectiveness of participatory approaches in the absence of support from researchers or the types of problems which teams may encounter [10]. An examination of these participatory approaches in practice is timely, given the ongoing and lively debates about the challenges and potential negative consequences of approaches such as co-production [19], which require more effort and resources than more traditional ‘top-down’ (e.g., theory-driven) interventions [10,11]. 

The present research considers the acceptability of an approach involving co-production and related methods to the commissioners of physical activity programmes and those who are responsible for designing such programmes. We evaluated the Greater Manchester Active Ageing (GM-AA) initiative—an innovative programme enacted across eight local authorities (Metropolitan Borough Councils (MBCs)) in the Greater Manchester area in England. The programme received £1 million from Sport England, over a two year period, with an explicit emphasis on trying ‘new ways’ of encouraging physical activity provision for older adults through increased participation of older adults. Each MBC had freedom to design their own programmes in response to local needs and capacities, but with an explicit criterion for funding to be used for one or more participatory approaches. 

The present research aimed to understand the experiences of service providers and decision-makers with these participatory approaches to developing interventions, and to comprehend barriers and facilitators to designing and implementing physical activity opportunities. 

## 2. Methods

### 2.1. Design and Setting

Semi-structured, qualitative interviews were conducted with MBC leads and stakeholders with key decision-making roles in the GM-AA Programme. Greater Manchester is a conurbation with a population of approximately 2.8 million people [20]. It is an area that includes areas of high deprivation on multiple indicators; eight of 10 Greater Manchester MBCs are ranked within the most deprived 100 of 317 local authorities in England, with six being within the 50 most deprived (by the average index of multiple deprivation score [21]). The GM-AA Programme started on 1 April 2018. Interviews took place approximately one year into this two-year project, when planning for local projects had been taking place (and some localities had commenced the implementation of projects), but before it was apparent how successful these participatory approaches were at increasing physical activity. 

### 2.2. Participants

Interview participants belonged to two groups. The first included representatives from MBCs who had decision-making roles in specifying the approach that each GM-AA project took, and/or were involved with securing GM-AA investment in their locality (‘MBC Lead’ participants). These individuals were therefore involved in deciding on the approach the locality projects took, but not necessarily involved in the on-the-ground delivery of new sessions. The second group included individuals from GM-wide stakeholder organisations who had contributed to the overall project through involvement in the initial bid to Sport England or through roles in commissioning and supporting the MBC applications (‘Stakeholder Organisation’ participants). Participants were purposively sampled to ensure that all participating MBCs were represented by at least one person, and there was representation from a range of stakeholder organisations (including Greater Sport, Sport England, and Greater Manchester Ageing Hub).

### 2.3. Data Collection

Semi-structured interviews took place between 18 December 2018 and 18 May 2019. Interviews were conducted face-to-face or by telephone and were structured using a topic guide (see Appendix A). The topics discussed included experiences of developing GM-AA projects, the effects of contextual factors on implementation, and what constitutes the successful provision of physical activity projects to older adults. The topic guides were used flexibly, with some topics covered in more depth with some individuals, according to their role in the process of commissioning and designing projects. Interviews were audio-recorded and transcribed verbatim. 

### 2.4. Analysis

Inductive thematic analysis was conducted with the aim of understanding the experiences of development and implementation from the perspective of study participants [22]. The Framework approach was used to structure the analysis [23]. Framework provides a transparent structure to the analysis process, which is particularly useful when multiple researchers are working with the dataset, and easily permits analysts to review the steps that other researchers have taken. The first and third authors familiarised themselves with the data by reading and re-reading interview transcripts and developed ‘codes’-labels that reflected important issues in the dataset. Codes were organised into a working thematic framework, including a list of categories and sub-categories. Two other authors (second and last authors) read transcript samples and reviewed and discussed the working thematic framework. The framework was applied to the full dataset (‘indexing’): This indicated where text within interviews fitted within the categories/ subcategories of the working thematic framework. Matrices were developed: Charts in which category contents were mapped by participants were produced so that researchers could compare category content across participants, as well as participant perspectives across categories (see Appendix A for an illustrative extract from a matrix). These matrices were interrogated to identify important and related issues in the dataset, and to generate insights into the issues considered. Through this process, initial categories of the working framework were further developed and refined to produce the final themes reported here.

## 3. Results

Twenty individuals were interviewed: 13 MBC Lead (MBC) and seven Stakeholder Organisation (SO) interviewees aged 20–59 years, with half aged 40–49 years. All self-identified as white; 16 were female and four were male. Most (19) interviews were conducted face-to-face and one was conducted by phone. Interviews ranged from 34 to 113 min (mean: 56 min). 

Three main themes which address the aims of this paper were identified (see Table 1): Experiences of participatory approaches; understanding of the acceptability of physical activity programmes by older adults; and resources and sustainability. The following report gives more space to findings that are novel, and less to findings that previous research has covered in detail.

### 3.1. Experiences of Participatory Approaches

#### 3.1.1. Experiences of Co-Production and Co-Design

MBC participants’ understanding of co-design and co-production varied widely. The description of ‘co-design’ by one MBC lead was more about gathering opinions, rather than having true involvement in developing physical activities: 

“We get them all in, give them like a tea or a coffee and some biscuits and get them chatting in a social element, and we do come round with a short questionnaire basically asking what activity they’d like to do on what day, what time, and just a rough idea of the barriers to the physical activity” (P6, MBC).

Other MBCs seemed to seek more in-depth input from older adults. For example, in the following quote, the ideas are coming from those in the community working with the providers in this case, rather than the providers developing the programme for the community without that local knowledge and input:

“People have got strengths, they’ve got assets and they’ve got some fantastic ideas, when we’ve sort of looked at numbers in the past and said, “Well, how would you get more inactive older people to come along?” They’ve come up with suggestions. They’ve really sort of taken ownership of the sessions. So, yeah, it’s happened very sort of organically” (P15, MBC).

Challenges that arose when attempting co-design approaches to programme development were discussed. Inexperience with co-design approaches meant that initial strategies were not always optimally effective. One MBC initially invited older adults to steering group meetings with an operational focus; this was not found to be an effective way to draw on older adults’ experience. An alternative approach, of organising separate meetings in a community setting, seemed more suitable for this MBC, enabling older adults to contribute to the decision-making process. 

There was a distinct sense that co-design approaches had important benefits for projects, as indicated in the quote from P15 above. In this locality, older adults were involved in every step, helping design the programme of activities, and with representatives attending steering group meetings.

#### 3.1.2. Experiences of Place-Based Working

Participants generally seemed to feel that a place-based approach meant looking at how projects could be embedded in the community. For example, one neighbourhood lacked leisure facilities and the MBC lead saw the GM-AA programme as an opportunity to develop ideas around finding alternative available resources: 

“It doesn’t have a specific leisure facility so therefore it gave us an added advantage of kind of testing other community place-based models, you know, what assets do we have in that community or that neighbourhood” (P7, MBC).

There was a perception that traditional leisure facilities, such as gyms, might be off-putting for older adults, and considering alternative venues could therefore be helpful for increasing participation: 

“It’s a lot about the facilities and where older adults would like to go. So if the provision’s in a hi-tech leisure centre the chance of getting older adults to want to engage in that, it’s sort of not understanding the provision of what the activity should be but where that activity should be based is massive” (P5, SO). 

One participant described a community centre within a park, which was a setting that was seen to have important benefits:

“I think one of the positives has been that it’s not a traditional kind of leisure centre setting. It’s very much, you know, green space park and then there’s an indoor space for people to go and meet and have a cup of tea. […] But it’s very kind of open and people feel comfortable there” (P15, MBC). 

Participants also considered the needs of different localities. In particular, as more deprived areas lacked resources and engagement opportunities, there was a particular concern about ensuring that older adults in those areas be included: 

“We are very aware that different localities have different levels of assets, community assets, community resources. That affects older people’s ability to engage in programmes like this. So if you are in an area that’s poorer or perhaps people are having to do paid work or have health problems or have carers’ responsibilities, not able to get out and about so much. So I know that some of the localities were trying to yeah targeting most deprived areas” (P1, SO).

#### 3.1.3. Partnership and Collaborative Working

Most participants seemed to consider older adults as assets, voicing an ethos of ‘doing with’ rather than ‘doing to’. Involving older adults in physical activity provision, and utilising their skills and connections, were seen as important ways to enhance projects and to maximise the programme reach: 

“Training older people to be trainers themselves is sustaining that model that we are looking to have, older people being assets, doing things for themselves, being in a good position to reach people within their own communities” (P1, SO). 

“They’ve come up with suggestions. So banners in the park, which have worked tremendously well, […] word-of-mouth, participants are putting leaflets in chip shops, you know, in libraries” (P15, MBC).

An expectation within the programme was that MBCs would develop projects in partnership with other key organisations, such as leisure providers and charities. Such collaboration was generally viewed positively, allowing MBCs to benefit from a wide range of experience: 

“I think if you work in partnership, you have all the plusses of the fact that your programme’s generally more successful. […] You’ve got that benefit of actually being able to learn from each other’s experience, having those contacts and connections in the community and higher up” (P13, MBC). 

Working collaboratively also seemed to have practical benefits, e.g., enabling quicker engagement with established older adult networks, and providing skills and experience to support projects: 

“For me, it’s a joint project, it’s a partnership […] And what they’ve been able to bring is access to all those [community] groups and their board members and the resident reps, […], it’s been a really positive experience” (P15, MBC).

However, it was acknowledged that it takes time to build relationships with partners, and that effective collaboration may depend on pre-existing relationships with organisations: 

“I think the most successful projects are, will be where you’ve got existing relationships, a lot of strong relationships at local level and trusted relationships. It’s very difficult to just to go in cold to an area and start things from scratch and to build up relationships, and confidence and trust” (P1, SO).

Miscommunication around expectations, capacity issues, and competition for limited financial resources were potential challenges for collaborating organisations, and could hinder optimal delivery:

“…it’s quite a barrier […], in that none of us have got much money. So people, if you’re not careful, are chasing money. And I think if it becomes about the money, then you’ve got a problem, because it should be about the programme and the older people locally” (P13, MBC). 

Despite the challenges, it was clear that partnership working could be beneficial in facilitating processes such as co-design and place-based working. 

### 3.2. Understanding of the Acceptability of Physical Activity Programmes by Older Adults

#### 3.2.1. Social Element 

Both MBC and stakeholder interviewees identified the social element of activities as highly important and key for participation: “You know, we’re selling activity but people are buying friendship” (P9, MBC). One interviewee noted how older adults would meet to socialise before or after the physical activity session: “So people were turning up early to have a brew, as well as staying at the end to have a brew” (P15, MBC). 

#### 3.2.2. Shifting the Norms around Physical Activity

Many interviewees saw changing how physical activity provision is thought of and spoken about to be central. It was felt that the way physical activity opportunities are traditionally described could have negative connotations, and that focusing on fun and pleasurable aspects of sessions would be beneficial: 

“People have negative perceptions of physical activity. And when we have the conversations we focus more on the, not the health messages or the physical activity messages but utilising the fun and the connections and getting out in the fresh air and the relaxation and things like mindfulness and things” (P11, MBC). 

It was recognised that systemic change is required to change long-standing ways of thinking and talking about physical activity, but that such change can be challenging when resources are limited: 

“And then you’re trying to change a system which has got embedded ways of working, which is financially under strain or stress and has a view of what older people are and do, you know. And you don’t have to wander around very often, very far, to look at the leaflets, the imagery, so on and so forth, that’s commonplace in leisure provision, to see that older people, you know, they’re not kind of part of the package at all, you know” (P18, SO).

#### 3.2.3. Accessibility 

Accessibility was seen as key for delivering successful programmes. A local venue, minimising travel, was considered important due to financial implications and psychological factors around travel, such as lacking confidence: 

“It’s very much around doorstep delivery as well because obviously transport and travel is an issue for lots of people including some older people, so it’s around making sure they’re in the right place, not just in terms of the usual inequalities but in terms of access generally” (P3, MBC).

A consideration of the physical environment of activity sessions to ensure that older adults felt at ease and the importance of social support in enabling engagement was discussed: 

“Because I think from the focus groups, a couple of the people said that actually they were quite fearful of walking in parks on their own, because they felt that people looked at them as though they were a bit strange and things like that, so I think it gives that real sort of, that bond, if you like, and makes people feel safer” (P19, MBC).

Access to marketing and promotional materials was raised by MBC leads. They proposed alternative marketing methods based on existing/developing relationships, or traditional approaches to publicity:

“…it’s getting that message out to them because the barrier is that a lot of them aren’t on social media, they don’t know how to access the information, so being in the area and on the ground and being that face of contact and going to where the older people are is a must” (P6, MBC).

However, one participant found that social media could be effective, and could engage younger individuals who can share information with older relatives. Encouragement from friends and family was seen to be important in facilitating engagement: 

“More often than not the wives really encourage the men, […] maybe it is tapping into the more active spouse, or that kind of thing, to get people through the door and like harness that friendship, that that need to make friends” (P9, MBC).

### 3.3. Resources and Sustainability

#### 3.3.1. Staffing and Timescales

Both stakeholder and MBC lead interviewees discussed the impact of staff turnover during the development or implementation of GM-AA projects. New staff joining projects partway through struggled to develop and deliver programs, feeling that they lacked background understanding: 

“I think with the change in management, one member of staff leaving and one coming off maternity leave, I think it has affected it a lot, because I think we were flowing really well” (P16, MBC).

Related to the staff capacity, time was considered a valuable resource, and tight timescales were found to be challenging. A central aspect of the GM-AA Programme was the expectation that MBCs would work with older adults and communities when developing projects. However, the timescales of the programme seemed to make such activities difficult:

“If you’re going to do it really true to the spirit of co-production and people and communities, it takes a really long time. And I think we had about three months from start to finish as I remember it. Well, that’s not time to engage with new people and communities and understand their lives and get them to help shape the plan” (P17, SO).

Effective engagement with communities took significant time as relationships and trust needed to be built. Where relationships with older adult communities do not already exist, connections need to be developed, and this may be challenging where timescales are tight and staff capacity is low, or impacted by turnover:

“Essentially it takes time and effort to hear older people’s voices and often setting up systems takes time and investment. And a lot of the local authorities do not have that older people’s network or forum–some do but a lot don’t. […] So it’s more difficult for somebody to go: right, here’s some money you can apply for, you need to involve older people. How do you get older people, you know? However if it’s already set up and you’ve already got older people telling you what it is they want, then that engagement is already there” (P1, SO).

#### 3.3.2. Sustainability

How projects might be sustained beyond the programme funding period was a concern for many interviewees, and there was a recognition that sustainability could be most challenging in areas of higher deprivation:

“Unfortunately, we’re still in a world where we’re on two or three year funding cycles and all of that, we all know that genuine long-term behaviour change takes time and it takes more time in places with less social capital and less, you know, to work with at the beginning. So, there will no doubt be a difference between the places, the more affluent places and the least affluent places in terms of actually impacting, and until we move to a world where we’re investing long-term, and we’re not on this project by project basis, yeah, it’s not ideal” (P17, SO).

Some locations were already aiming to ensure the sustainability of activities by charging a small fee to participants, although this raised the concern that such an approach could make it difficult for older adults with limited financial resources to attend: 

“The cost associated helps to pay for the instructor long term and the venue hire, without that cost it’s just not sustainable. So if in deprived areas people couldn’t afford to do that it would affect the sustainability” (P6, MBC).

The sense that older adults might be viewed as assets was supported by one interviewee, who perceived local older adults to be potentially more valuable than staff in facilitating project sustainability:

“If you’ve got somebody from a similar age […] grown up in the same area, who knows the language, who knows some of the social networks, who knows some of the families who live in the place and what their concerns are […] you’re going to have more impact and those people stay in the community, you know. They don’t then go and get another job two years later” (P18, SO).

This stakeholder felt that older adults delivering physical activity sessions themselves was a powerful model because participants might feel that they can relate to the deliverer, upskilling older adults from the community may mean that the skills and delivery are more likely to stay in the community than if they are delivered by externally commissioned staff, and the individual might be less likely to leave a project on cessation of funding. 

One MBC was already following a model of training individuals from the community to deliver activity sessions: 

“Yeah, say for example if we want chair-based activities and we recognise there aren’t many available chair-based deliverers […], then what we start to do more and more now is upskill an individual from within the community or even a participant that wants to be involved so we’ve got that legacy left for the programme” (P7, MBC).

These findings would suggest that utilising key aspects of participatory approaches in the GM-AA Programme may not only support the design and delivery of acceptable projects with which older adults will engage, but also help projects to be sustainable. 

## 4. Discussion

Interviewees’ perspectives of the new approaches were generally positive: The approach was not only seen as useful, but also valuable, with partnerships formed and benefits experienced that could inform subsequent working. Older adults were viewed as assets within co-design and place-based approaches, with value seen in them contributing to and taking ownership of projects. Benefits were also seen in increased working with partner organisations. Interviewees felt that the language used when talking about and promoting physical activity needed to change in an effort to highlight fun and social aspects. Challenges to carrying out co-design and co-production were also identified. Developing working relationships with partner organisations was recognised as requiring a significant period of time to do well, particularly where there were no pre-existing partner relationships or there was inexperience in using these approaches. High staff turnover and tight timelines further exacerbated these issues. Sustainability beyond the period of funding was a key concern for interviewees, particularly for areas of high deprivation. Viewing older adults as assets enabled interviewees to see them as part of the solution to the continuation of projects once funding ceases. 

It was apparent from descriptions of activities engaged in that not all of the localities carried out true co-production and co-design with older adults, but sometimes referred to consultation work. The Social Care Institute for Excellence (SCIE) [24] identified a number of barriers to co-production, including a lack of knowledge and understanding of what is involved, as well as time pressures and shortages of funding. In line with this, interviewees in the current study mentioned a lack of time as a key reason for why true co-production might not have occurred. Some interviewees mentioned that involving older adults in steering group meetings did not work well. One study looking at the experiences of health professionals and peer leaders in working together found that tensions arose when peer leaders felt they were not given status or a strong voice in their role, indicating the need for a ‘culture of mutual respect’ [25]. 

Interviewees discussing a place-based approach to working saw this as an opportunity to determine what physical assets are already in the community, and how ideas could be developed around these assets. It was felt that community venues (rather than leisure facilities) could be more appealing to inactive older adults, and accessibility was seen as important. A review of studies examining interventions to promote physical activity in older adults found that the environment in which physical activity sessions are provided is important to older adults, with participation at least in part depending on the availability and proximity of environments perceived as attractive, safe, and low-cost [26].

The concept of older adults being assets came across strongly during discussions of co-production and place-based working. An example of a successful model utilising the skills of older adults in supporting engagement with physical activity is the Someone Like Me programme, which involves older adult peer mentors supporting other older adults in physical activity [27]. Support for peer volunteers increasing physical activity levels in older adults has been found [28,29].

This study took an in-depth look at how individuals with development and decision-making roles experienced and perceived developing projects using participatory approaches. However, there are other important perspectives to be taken on board: It is also important to understand the experiences of the older adults who take part in the projects developed and the perceptions of the individuals who deliver sessions. 

This study evaluated the experiences within a single programme in the UK. Other regions could have different organisational structures in place, and cultural differences could impact the experiences of such participatory approaches, so it will be important to evaluate similar programmes developed in other locations. However, the present study involved representatives of eight MBCs in a conurbation with a combined population of 2.8 million people. Furthermore, the MBCs included a number of the most deprived locations in England, with corresponding pressures on financial and time resources. Given this potentially challenging environment for developing new approaches, the positive experiences reported here should be possible to replicate in less deprived areas of the UK and internationally. It is also important to note that the participatory approaches covered by the present report did not include the substantial involvement of researchers as participants: This is unusual in reports on the co-production of interventions [10,18]. 

The main finding of the present study is that it shows the feasibility of using novel participatory approaches by people who have a limited experience of these, and without a good deal of support from researchers or experts in these approaches. Furthermore, the participatory approaches to physical activity examined in this paper seemed to yield important benefits in the development of projects that are suitable for the project location and acceptable to older adult participants. The major implication of this research is that, even in a difficult financial environment with a deprived population, the experiences across multiple MBCs and organisations were generally positive, with all partners seeing value in these participatory approaches compared to traditional approaches to increasing physical activity. This new way of working also appeared to bring about a new way of thinking and speaking about older adults and physical activity, with participants talking about an ethos of ‘doing with’ rather than ‘doing to’. Valuing older adults’ views and involving them in processes brought about ideas and feedback that ensured projects were acceptable and appealing, with an emphasis on social benefits. 

However, where organisations are expected to implement these new approaches, it is important that they are provided with sufficient periods of time and appropriate staffing to build trusting relationships with older adults and partner organisations. They also need to receive appropriate training and support. 

Ensuring the sustainability of programmes is a key concern. An important benefit of the co-design approach was that older adults were able to consider sustainability and contribute recommendations for achieving this. Taking a place-based approach seemed to help identify assets that were already in place, independently of funding. However, there is a need to take a long-term approach to investing in physical activity provision for older adults to ensure that the creative and engaging projects developed in a programme such as this are able to be sustained and prosper beyond a short-term funding cycle.

## 5. Conclusions

In sum, incorporating participatory approaches, such as co-design, co-production and a place-based approach were seen to yield important benefits by individuals involved in designing and making decisions around physical activity provision for older adults. Sufficient funding to ensure adequate staffing, support for staff, and sustainability of projects beyond a short-term funding cycle is required.

## Figures and Tables

**Table 1 ijerph-18-02172-t001:** Summary of themes and sub-themes.

Theme	Sub-Themes
Experiences of participatory approaches	Experiences of co-production and co-design Experiences of place-based working Partnership and collaborative working
Understanding of acceptability of physical activity programmes by older adults	Social element Shifting the norms around physical activity Accessibility
Resources and sustainability	Staffing and timescales Sustainability

## Data Availability

The anonymised dataset analysed during the current study is available from the corresponding author on reasonable request. Please note that due to the roles of the individuals interviewees, individuals could be identified from their transcripts, despite attempts being made to anonymise the transcripts. For this reason, we do not want these transcripts to be in the public domain.

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
