# Peer review of "How Do Decision Makers and Service Providers Experience Participatory Approaches to Developing and Implementing Physical Activity Interventions with Older Adults? A Thematic Analysis"

_ijerph, 2021, doi:10.3390/ijerph18042172_

Round 1

Reviewer 1 Report

The authors note that there are few evaluations of success with engaging older adults in physical activity. This paper does not identify the efficacy of the approaches used in this study. Interviews were conducted related to the process, not the results.

Author Response

The authors note that there are few evaluations of success with engaging older adults in physical activity. This paper does not identify the efficacy of the approaches used in this study. Interviews were conducted related to the process, not the results.

We completely agree with this comment.  We have now clarified the abstract conclusions on exactly this point on lines 42-47, where we could previously have been interpreted as commenting on effectiveness or efficacy, as follows:

Conclusions: Incorporating participatory approaches were perceived to yield important benefits.  Interviewees highlighted that to ensure success, sufficient time needs to be provided to develop good working relationships with older adults and partner organisations.  They also emphasized that sufficient funding to ensure adequate staffing and sustainability of projects is required to allow benefits to be gained.  Importantly, implementation of these approaches appears feasible across a range of local authorities.

Reviewer 2 Report

1. The aims should more clearly identify what "new ways of working" are being explored. The introduction mentions co-production, co-design, place-based approaches, and asset-based approaches, but it is not clear which were used in the design of the GM-AA program or evaluated in this analysis. The conclusion similarly mentions 'the novel approach to physical activity intervention development" (line 443) which is not explained in this paper. 

2. Similarly, the introduction of this manuscript makes a point of defining co-production and co-design, but they seemed to be used interchangeably? Were participants aware of the difference? Did the intervention use both approaches? I would suggest either sticking with one term (whatever one is most relevant to this intervention). Similarly, how were place-based approaches used in the intervention?

2. line 68-70 - 'introducing' is used twice in this sentence. 

3. line 84-86 - this sentence could also use revision, perhaps replacing the first 'physical activities' 

4. It would be helpful to know more about the roles of each participant in relation to the intervention. I find the descriptions provided - MBC lead and stakeholder organizations - to be vague and I am not clear what role the research participants played in the co-design/co-development process.

6. Line 401-402 indicates some evaluation of the amount of co-design or co-production? How was this assessed? It was not part of the original aims.

7. The manuscripts switches between 'interviewees' and 'participants' I found this a little confusing since 'participants' is also used in reference to the older adult participants in the physical activity intervention

Author Response

  1. The aims should more clearly identify what "new ways of working" are being explored. The introduction mentions co-production, co-design, place-based approaches, and asset-based approaches, but it is not clear which were used in the design of the GM-AA program or evaluated in this analysis. The conclusion similarly mentions 'the novel approach to physical activity intervention development" (line 443) which is not explained in this paper. 

We have now clarified the aims in this regard.  First, we now use the phrase “participatory approaches” instead of “new ways of working” throughout, as this new phrase this more clearly identifies what these various approaches have in common.  We have also made clearer that this is an umbrella term that covers the various approaches listed on lines 70-77:

There is increasing interest in participatory approaches to promote physical activity [8].  These approaches centre around increasing participation of older adults in the development of physical activity interventions, which are valued in the locations they are implemented, in order to increase physical activity. Such approaches typically aim to embed long-term, sustainable physical activity programmes within the neighbourhoods where older adults live, rather than being ‘interventions’ delivered for a fixed period of time and then withdrawn.

The present research considers various participatory approaches to involving older adults…

Second, we now more explicitly state that the MBC Leads in each locality were responsibility for deciding which of these various participatory approaches were employed, on lines 114-124.  Often the MBC Leads employed more than one of these approaches.  So in sum, the evaluation is of the usage of a toolkit of participatory approaches.

The present research considers the acceptability of an approach involving co-production and related methods, to the commissioners of physical activity programmes and those who are responsible for designing such programmes. We evaluated the Greater Manchester Active Ageing (GM-AA) initiative, an innovative programme enacted across eight local authorities (Metropolitan Borough Councils [MBCs]) in the Greater Manchester area in England. The programme received £1 million from Sport England, over a two year period, with an explicit emphasis on trying ‘new ways’ of encouraging physical activity provision for older adults through increased participation of older adults.  Each MBC had freedom to design their own programmes in response to local needs and capacities, but with an explicit criterion for funding to be the use of one or more participatory approaches.

We similarly have revised the Discussion sentence in lines 491-493 with this clearer explanation.

Further, the participatory approaches to physical activity examined in this paper seemed to yield important benefits in the development of projects that are suitable for the project location and acceptable to older adult participants. 

  1. Similarly, the introduction of this manuscript makes a point of defining co-production and co-design, but they seemed to be used interchangeably? Were participants aware of the difference? Did the intervention use both approaches? I would suggest either sticking with one term (whatever one is most relevant to this intervention). Similarly, how were place-based approaches used in the intervention?

We are now clearer on lines 114-124 that the MBCs were able to choose whichever participatory approach they thought was most useful for the development of their interventions:

The present research considers the acceptability of an approach involving co-production and related methods, to the commissioners of physical activity programmes and those who are responsible for designing such programmes. We evaluated the Greater Manchester Active Ageing (GM-AA) initiative, an innovative programme enacted across eight local authorities (Metropolitan Borough Councils [MBCs]) in the Greater Manchester area in England. The programme received £1 million from Sport England, over a two year period, with an explicit emphasis on trying ‘new ways’ of encouraging physical activity provision for older adults through increased participation of older adults.  Each MBC had freedom to design their own programmes in response to local needs and capacities, but with an explicit criterion for funding to be the use of one or more participatory approaches.

We now use the term “participatory approaches” as an umbrella term throughout.  One of the first findings we report on lines 202-217 shows that understanding of these terms varied between MBCs:

MBC participants’ understanding of co-design and co-production varied widely.  The description of ‘co-design’ by one MBC lead was more about gathering opinions, rather than having true involvement in developing physical activities:

“We get them all in, give them like a tea or a coffee and some biscuits and get them chatting in a social element, and we do come round with a short questionnaire basically asking what activity they’d like to do on what day, what time, and just a rough idea of the barriers to the physical activity” (P6, MBC).

“People have got strengths, they've got assets and they've got some fantastic ideas, when we've sort of looked at numbers in the past and said, "Well, how would you get more inactive older people to come along?"  They’ve come up with suggestions. They've really sort of taken ownership of the sessions.  So, yeah, it's happened very sort of organically”  (P15, MBC).

  1. line 68-70 - 'introducing' is used twice in this sentence. 

We have revised the second sentence to remove repetitive wording, as follows:

There is increasing interest in participatory approaches to promote physical activity [8].  These approaches centre around promoting participation of older adults in the development of physical activity interventions

  1. line 84-86 - this sentence could also use revision, perhaps replacing the first 'physical activities' 

We have revised this sentence to remove repetition on lines 70-77:

There is increasing interest in participatory approaches to promote physical activity [8].  These approaches centre around promoting participation of older adults in the development of physical activity interventions, which are valued in the locations they are implemented, in order to increase physical activity. Such approaches typically aim to embed long-term, sustainable physical activity programmes within the neighbourhoods where older adults live, rather than being ‘interventions’ delivered for a fixed period of time and then withdrawn.

  1. It would be helpful to know more about the roles of each participant in relation to the intervention. I find the descriptions provided - MBC lead and stakeholder organizations - to be vague and I am not clear what role the research participants played in the co-design/co-development process.

We cannot give a detailed description of which role each specific individual played as given the nature of the interviewees that would allow individuals to be identified, which would breach our ethical clearance.  However, we now include more detailed description on lines 142-154 of how people in each of these two roles played a role in the development of these participatory approaches:

Participants

Interview participants belonged to two groups: (a) representatives from MBCs who had decision-making roles in specifying the approach that each GM-AA project took, and/or were involved with securing GM-AA investment in their locality (‘MBC Lead’ participants).  These individuals were therefore involved in deciding on the approach the locality projects took, but not necessarily involved in the on-the-ground delivery of new sessions; (b) individuals from GM-wide stakeholder organisations who had input into the overall project through involvement in the initial bid to Sport England or through roles in commissioning and supporting the MBC applications (‘Stakeholder Organisation’ participants).  Participants were purposively sampled to ensure all participating MBCs were represented by at least one person, and there was representation from a range of stakeholder organisations (including Greater Sport, Sport England and Greater Manchester Ageing Hub).

  1. Line 401-402 indicates some evaluation of the amount of co-design or co-production? How was this assessed? It was not part of the original aims.

We did not directly evaluate the amount of co-design or co-production, as this was a qualitative study and quantification would not be appropriate.  We have now revised this sentence so that is makes it clearer that the evidence for the lack of true co-production/ co-design was provided by how participants talked about what it was they did.  We link this observation back more directly to our findings on lines 446-448:

It was apparent from descriptions of activities engaged in that not all the localities carried out true co-production and co-design with older adults, but sometimes referred to consultation work.

  1. The manuscripts switches between 'interviewees' and 'participants' I found this a little confusing since 'participants' is also used in reference to the older adult participants in the physical activity intervention

We now use “interviewees” more consistently throughout.

Reviewer 3 Report

Thank you for the opportunity to review your work.

Please see below comments/feedback regarding your study.

line 68-89: this paragraph needs to be incorporated with the next paragraph.

Line 95: is this the present approach? I don't see this method from the list above which includes co-production, co-design, place-based, and asset.
If this is a core keyword of this research, it is necessary to have some research summary regarding this including definition, research trend in older adults study.

Line 130-133: is this group considered as a "service provider"? If not, you need to change the abstract from "service provider" to "stakeholder"?

Lines 150-154: it might not be necessary to describe the roles of each author. Instead of the roles/expertise of the author, it would be necessary to provide an analysis process within the methodological framework.

Results chapter: Entire Results chapter needs revision since the reader might not be interested in what the participants said but how you processed the conversation based on the analysis framework.

Conclusion chapter: Please consider splitting this chapter into two: Discussion and Conclusion.

Author Response

Thank you for the opportunity to review your work.  Please see below comments/feedback regarding your study.

  1. line 68-89: this paragraph needs to be incorporated with the next paragraph.

This is a helpful comment: we have now re-ordered the material in what were previously paragraphs three and four in new lines 70-91, which reduces redundancy and helps flow:

There is increasing interest in participatory approaches to promote physical activity [8].  These approaches centre around promoting participation of older adults in the development of physical activity interventions, which are valued in the locations they are implemented, in order to increase physical activity. Such approaches typically aim to embed long-term, sustainable physical activity programmes within the neighbourhoods where older adults live, rather than being ‘interventions’ delivered for a fixed period of time and then withdrawn.

The present research considers various participatory approaches to involving older adults, including co-production, co-design, place-based working and an asset-based approach.  There is a variety of definitions for these approaches that derive from different disciplinary backgrounds [9], leading to frequent areas of disagreement [10,11].  In the present research, terms are used as follows: Co-production is an umbrella term for activities that aim to fully involve end-users in the development of interventions, by viewing the experiential knowledge of these end-users as core to the success of their development [10]. A related concept, co-design, emphasises involvement in identifying the problem and how to go about addressing it, rather than involvement in the development or delivery of interventions [12]. A place-based approach considers both local needs and local assets [13], drawing on older people’s extensive knowledge of the communities and environments in which they live. This relates to taking an asset-based approach, where the experiences and skills of older adults are recognised and valued, as are the resources available in a local area [14]. 

  1. Line 95: is this the present approach? I don't see this method from the list above which includes co-production, co-design, place-based, and asset.
    If this is a core keyword of this research, it is necessary to have some research summary regarding this including definition, research trend in older adults study.

We now use the phrase “participatory approaches” instead of “new ways of working” as an umbrella term throughout, as this new phrase this more clearly identifies what these various approaches have in common.  We have been explicit about how this is an umbrella term that encompasses the terms mentioned in the introduction on lines 70-79:

There is increasing interest in participatory approaches to promote physical activity [8].  These approaches centre around promoting participation of older adults in the development of physical activity interventions, which are valued in the locations they are implemented, in order to increase physical activity. Such approaches typically aim to embed long-term, sustainable physical activity programmes within the neighbourhoods where older adults live, rather than being ‘interventions’ delivered for a fixed period of time and then withdrawn.

The present research considers various participatory approaches to involving older adults, including co-production, co-design, …

  1. Line 130-133: is this group considered as a "service provider"? If not, you need to change the abstract from "service provider" to "stakeholder"?

We have revised the abstract on lines 21-33 to better match the text describing the two groups of participants:

Background: Physical activity confers numerous health and well-being benefits on older adults, but many older adults are inactive.  Interventions designed to increase physical activity in older adults have typically produced only small effects and have not achieved long-term changes. There is increasing interest in participatory approaches to promote physical activity, such as co-production, co-design and place-based approaches, but they have typically involved researchers as participants.  This study aimed to understand the experiences of decision-makers and service developers with the introduction of such participatory approaches when developing new physical activity programmes outside of a research setting. Methods: Semi-structured, qualitative interviews were conducted with 20 individuals who were involved in commissioning or developing the Greater Manchester Active Ageing Programme.  This programme involved funding eight local authorities within Greater Manchester, England, to produce physical activity projects for older adults, involving participatory approaches.  An inductive thematic analysis was conducted, structured using the Framework approach.

  1. Lines 150-154: it might not be necessary to describe the roles of each author. Instead of the roles/expertise of the author, it would be necessary to provide an analysis process within the methodological framework.

We have provided more details of the analysis process that we followed on lines 164-184, namely employing thematic analysis using the framework approach.  We continue to describe the roles of each author, as this allows the reader to see exactly who conducted each aspect of the research, an approach that is usual within qualitative research.

Analysis

Inductive thematic analysis was conducted with the aim of understanding experiences of development and implementation from the perspective of study participants [20]. The Framework approach was used to structure the analysis [21]. Framework provides a transparent structure to the analysis process, which is particularly useful when multiple researchers are working with the dataset, and easily permits analysts to review the steps other researchers have taken.  The first and third authors familiarized themselves with the data by reading and re-reading interview transcripts and developed ‘codes’ – labels that reflected important issues in the dataset.  Codes were organized into a working thematic framework – a list of categories and sub-categories.  Two other authors (second and last authors) read transcript samples and reviewed and discussed the working thematic framework.  The framework was applied to the full dataset (‘indexing’): this indicated where text within interviews fitted within the categories/ subcategories of the working thematic framework.  Matrices were developed: charts in which category content were mapped by participant such that researchers could compare category content across participants, and also participant perspectives across categories.  These matrices were interrogated to identify important and related issues in the dataset, and to generate insights into the issues considered.  Through this process, initial categories of the working framework were further developed and refined into the final themes reported here.

  1. Results chapter: Entire Results chapter needs revision since the reader might not be interested in what the participants said but how you processed the conversation based on the analysis framework.

We report illustrative quotations in the results chapter, which is standard practice for studies using qualitative methods because the quotations provide evidence for our assertions/ conclusions (in the way that tables and graphs provide evidence in quantitative studies).  Indeed it is nearly unheard of to not include such quotations.  The products of the analysis are themes and sub-themes, which again is standard practice in qualitative research.

  1. Conclusion chapter: Please consider splitting this chapter into two: Discussion and Conclusion.

We now label this section Discussion, with “conclusion” as a final sub-heading.

Reviewer 4 Report

Thank you for the opportunity to review this manuscript, which provides an interesting study of the physical activity perceptions of decision makers, as they relate to older adult physical activity.  Given the older adult population is the least active age group, effective strategies are needed for this population, and this study attempts to provide some observational details on potential ways to target physical activity increases, by involving older adults in the design and production of strategies for activity promotion.

Below are some comments to address:

In the introduction, can you perhaps add some background on the effectiveness of service providers to promote/encourage physical activity in older adults? Is there evidence this works?

Methods: By “high deprivation”, does that mean socially disadvantaged? Or low-income? You might clarify that in this section

Was the interview guide followed exactly the same for each participant (i.e. the fidelity)? You might report on that.

Results: Were only three themes identified from all of the interviews, or were those just the main ones? Given the description analyses matrices, and the way the results are displayed, it seems there may have been 3 main themes, with multiple sub-themes or categories in each. Could you perhaps provide a display of that as a table or figure? I think visually showing the matrices would enhance this paper and better inform others of the thematic analysis

Line 178: Can you provide an example of the more in-depth input?

Discussion: While reading this, I kept waiting for a big takeaway message of this study, which is slightly lacking. Can you add that somewhere? What is the most important finding(s) based on this study? Comparison to the literature was adequate, but readers may better need to know how to use the results

Does reference #19 need the word [internet] in there? Should health be capitalized in reference #13? Should the hyperlink be removed in reference #22?

Author Response

Thank you for the opportunity to review this manuscript, which provides an interesting study of the physical activity perceptions of decision makers, as they relate to older adult physical activity.  Given the older adult population is the least active age group, effective strategies are needed for this population, and this study attempts to provide some observational details on potential ways to target physical activity increases, by involving older adults in the design and production of strategies for activity promotion.

Below are some comments to address:

  1. In the introduction, can you perhaps add some background on the effectiveness of service providers to promote/encourage physical activity in older adults? Is there evidence this works?

There is not much evidence on this point, however we now cite the preliminary evidence that is available on lines 98-113.  We aim to clarify that there is still not compelling evidence that this approach is effective, which is why the interest in these new participatory approaches.  We now frame the aims of our study in the initial evidence, and lack of a definitive answer to this question.

Despite growing interest in using these new approaches, there are few evaluations of their success, especially in relation to groups such as older adults [15]. In particular, there is limited evidence concerning the effectiveness and impact of participatory approaches with older adults [15].  There has been some evaluation of such programmes, aimed at older adults and adults with health conditions, delivered in partnerships by local government, local NHS organisations and voluntary sector organisations. These suggest increased levels of physical activity in those who persist with the programmes, as well as improved mental wellbeing, with some projects continuing beyond the funding period [28,29].  Further, many existing co-production exercises to date have involved researchers as a key participant group alongside service providers and service users [16].  There is limited knowledge about the effectiveness of participatory approaches in the absence of support from researchers or of the types of problems which teams may encounter [10].  Examination of these participatory approaches in practice is timely, given the ongoing and lively debates about the challenges and potential negative consequences of approaches such as co-production [17], which do require more effort and resources than more traditional ‘top-down’ (e.g. theory-driven) interventions [10,11].

  1. Methods: By “high deprivation”, does that mean socially disadvantaged? Or low-income? You might clarify that in this section

We are now clearer on lines 133-136 that we have operationalised deprivation in terms of the index of multiple deprivation, which is a geographically based index, including elements including housing quality, education, crime and various other elements:

It is an area that includes areas of high deprivation on multiple indicators; eight of 10 Greater Manchester MBCs are ranked within the most deprived 100 of 317 local authorities in England, six being within the 50 most deprived (by average index of multiple deprivation score, [19]).

  1. Was the interview guide followed exactly the same for each participant (i.e. the fidelity)? You might report on that.

We used the interview guide flexibly, i.e. it was a guide of issues that the interview should cover, rather than an inflexible inventory.  Some topics were covered in more depth for each participant than for other participants.  We are now clearer in our description of this point on lines 160-162.

Data collection

Semi-structured interviews took place between 18th December 2018 and 18th May 2019.  Interviews were conducted face-to-face or by telephone and were structured using a topic guide (see Additional File 1).  Topics discussed included experiences of developing GM-AA projects, effects of contextual factors on implementation, and what constitutes successful provision of physical activity projects to older adults.  The topic guides were used flexibly, with some topics covered in more depth with some individuals, according to their role in the process of commissioning and designing projects.  Interviews were audio-recorded and transcribed verbatim. 

  1. Results: Were only three themes identified from all of the interviews, or were those just the main ones? Given the description analyses matrices, and the way the results are displayed, it seems there may have been 3 main themes, with multiple sub-themes or categories in each. Could you perhaps provide a display of that as a table or figure? I think visually showing the matrices would enhance this paper and better inform others of the thematic analysis

Interviews typically covered a range of issues, not all of which were relevant for addressing the aims of this paper.  We identified three key aims which address the research aims, and within these we gave more space to novel/interesting findings than to issues previous research has covered in detail.  We are now clear on lines 191-195 that three main themes were identified from the interviews, and that that we focussed space in the present report on the more novel/ interesting findings:

Three main themes were identified which addressed the aims of this paper (see Table 1): experiences of participatory approaches; understanding of acceptability of physical activity programmes to older adults; and resources and sustainability.  The following report gives more space to findings that are novel, and less to findings that previous research has covered in detail.

We now provide a table on line 198 summarizing the three themes and sub-themes to assist readers in following our findings

Table 1: Summary of themes and sub-themes

Theme

Sub-themes

Experiences of participatory approaches

·         Experiences of co-production and co-design

·         Experiences of place-based working

·         Partnership and collaborative working

Understanding of acceptability of physical activity programmes to older adults

·         Social element

·         Shifting the norms around physical activity

·         Accessibility

Resources and sustainability

·         Staffing and timescales

·         Sustainability

We do not feel it is appropriate to include the full matrices as providing the full details of what participants said about various topics could enable identification of individual participants (which would go against consent provided and our ethical approval).  However, we agree that seeing an example of a matrix might help readers to understand what this stage of the analysis process entailed.   Therefore, as an example, we are now including a selective, illustrative extract of a matrix as an Additional File, which we flag up on line 180.

  1. Line 178: Can you provide an example of the more in-depth input?

We now provide an example of this on lines 209-217:

“People have got strengths, they've got assets and they've got some fantastic ideas, when we've sort of looked at numbers in the past and said, "Well, how would you get more inactive older people to come along?"  They’ve come up with suggestions. They've really sort of taken ownership of the sessions.  So, yeah, it's happened very sort of organically”  (P15, MBC).

  1. Discussion: While reading this, I kept waiting for a big takeaway message of this study, which is slightly lacking. Can you add that somewhere? What is the most important finding(s) based on this study? Comparison to the literature was adequate, but readers may better need to know how to use the results

This suggestion is useful: we now attempt to be clearer about the implications of this research, without overstating the implications of the present research regarding effectiveness of this approach.  Specifically we have added the following to the first paragraph of the Discussion on lines 431-433:

Interviewees’ perspectives of the new approaches were generally positive: the approach was not only seen as useful, but also valuable with partnerships formed and benefits experienced that could inform subsequent working. 

We also include a clearer presentation of the implications in the Discussion on lines 489-502:

The main finding is that it shows the feasibility of using novel participatory approaches by people who had limited experience of these, and without a good deal of support from researchers or experts in these approaches.  Further, the participatory approaches to physical activity examined in this paper seemed to yield important benefits in the development of projects that are suitable for the project location and acceptable to older adult participants.  The major implication of this research is that, even in a difficult financial environment with a deprived population, the experiences across multiple MBCs and organisations were generally positive, with all partners seeing value in these participatory approaches compared to traditional approaches to increasing physical activity.  This new way of working also appeared to bring about a new way of thinking and speaking about older adults and physical activity, with participants talking about an ethos of ‘doing with’ rather than ‘doing to’. Valuing older adults’ views and involving them in processes brought about ideas and feedback that ensured projects were acceptable and appealing, with an emphasis on social benefits.

  1. Does reference #19 need the word [internet] in there? Should health be capitalized in reference #13? Should the hyperlink be removed in reference #22?

We have revised the references in light of these helpful observations.

Round 2

Reviewer 2 Report

This is a comprehensive revision and the authors have addressed all of my concerns.

Author Response

We thank the reviewer for their positive comments

Reviewer 3 Report

  1. Results chapter: Entire Results chapter needs revision since the reader might not be interested in what the participants said but how you processed the conversation based on the analysis framework.

We report illustrative quotations in the results chapter, which is standard practice for studies using qualitative methods because the quotations provide evidence for our assertions/ conclusions (in the way that tables and graphs provide evidence in quantitative studies).  Indeed it is nearly unheard of to not include such quotations.  The products of the analysis are themes and sub-themes, which again is standard practice in qualitative research.

-> I understand this is a study based on the qualitative method. But the results should not be just presenting what your team has received from interviewees. The results should show what you have identified from the methods you adopted, but this results chapter is a collection of the responses.

  1. Conclusion chapter: Please consider splitting this chapter into two: Discussion and Conclusion.

We now label this section Discussion, with “conclusion” as a final sub-heading

--> I don't see the conclusion.

Author Response

5. Results chapter: Entire Results chapter needs revision since the reader might not be interested in what the participants said but how you processed the conversation based on the analysis framework.

 We report illustrative quotations in the results chapter, which is standard practice for studies using qualitative methods because the quotations provide evidence for our assertions/ conclusions (in the way that tables and graphs provide evidence in quantitative studies).  Indeed it is nearly unheard of to not include such quotations.  The products of the analysis are themes and sub-themes, which again is standard practice in qualitative research. 

-> I understand this is a study based on the qualitative method. But the results should not be just presenting what your team has received from interviewees. The results should show what you have identified from the methods you adopted, but this results chapter is a collection of the responses. 

We agree with this reviewer that one should not just present what was elicited from interviewees.  That is, the findings should be the result of careful, rigorous collation, interpretation and analysis of the entire dataset.  For this reason, we present interpretation and analysis for all themes, using quotations to evidence our interpretation and analysis.  For example in the first sub-theme, "Experiences of co-production and co-design", it can be seen that quotations make up two out of the six paragraphs of our paper, with the other four paragraphs making more general statements about how participants viewed co-design and co-production:

MBC participants’ understanding of co-design and co-production varied widely. The description of ‘co-design’ by one MBC lead was more about gathering opinions, rather than having true involvement in developing physical activities:

“We get them all in, give them like a tea or a coffee and some biscuits and get them chatting in a social element, and we do come round with a short questionnaire basically asking what activity they’d like to do on what day, what time, and just a rough idea of the barriers to the physical activity” (P6, MBC).

Other MBCs seemed to seek more in-depth input from older adults. For example, in the following quote the ideas are coming from those in the community working with the providers in this case, rather than the providers developing the programme for the community without that local knowledge and input::

“People have got strengths, they've got assets and they've got some fantastic ideas, when we've sort of looked at numbers in the past and said, "Well, how would you get more inactive older people to come along?" They’ve come up with suggestions. They've really sort of taken ownership of the sessions. So, yeah, it's happened very sort of organically” (P15, MBC).

Challenges that arose when attempting co-design approaches to programme development were discussed. Inexperience with co-design approaches meant that initial strategies were not always optimally effective. One MBC initially invited older adults to steering group meetings with an operational focus; this was not found to be an effective way to draw on older adults’ experience. An alternative approach, organising separate meetings in a community setting, seemed more suitable for this MBC, enabling older adults to contribute to the decision-making process.

There was a distinct sense that co-design approaches had important benefits for projects as indicated in the quote from P15 above. In this locality, older adults were involved in every step, helping design the programme of activities, and with representatives at-tending steering group meetings.

  1. Conclusion chapter: Please consider splitting this chapter into two: Discussion and Conclusion.

We now label this section Discussion, with “conclusion” as a final sub-heading

--> I don't see the conclusion.

We now include the heading "conclusion" on line 498

Reviewer 4 Report

Thank you for the opportunity to review this revised manuscript. The authors have addressed all of my previous comments, which has much improved the manuscript. I appreciate the authors’ efforts to address those comments and feel the addition of the themes/sub-themes table helps the readability of the results. I have no additional comments.  

Author Response

We thank this reviewer for their positive feedback